# Evolutionary Shift of Insect Diapause Strategy in a Warming Climate: An Intra-Population Evidence from Asian Corn Borer

**DOI:** 10.3390/biology12060762

**Published:** 2023-05-24

**Authors:** Lianxia Wang, Kaiqiang Liu, Xiumei Zhao, Tiantao Zhang, Ming Yuan, Kanglai He

**Affiliations:** 1Qiqihar Branch of Heilongjiang Academy of Agricultural Sciences, Qiqihar 161000, China; 2State Key Laboratory for the Biology of the Plant Diseases and Insect Pests, Institute of Plant Protection, Chinese Academy of Agricultural Sciences, Beijing 100193, China

**Keywords:** *Ostrinia furnacalis*, voltinism, diapause, model, climate warming

## Abstract

**Simple Summary:**

An evolutionary shift in insect diapause strategy has been revealed in a historically univoltine population of *Ostrinia furnacalis*, a commonly facultative diapause species. Warmer climates are likely to stimulate facultative individuals in the population to shift from univoltinism to bivoltinism due to the compensatory effect of elevated temperature. Climate warming has driven the population to evolve toward dominantly bivoltine or even trivoltine, further leading to more severe damage. To accurately predict phenology and population dynamics in *O. furnacalis*, it is imperative to account for both the proportions of different diapause individuals with the population and temperature.

**Abstract:**

Herbivorous insects having variable numbers of generations annually depending on climate and day length conditions are increasingly breeding additional generations driven by elevated temperature under the scenario of global warming, which will increase insect abundance and result in more frequent damage events. Theoretically, this relies on two premises, i.e., either an evolutionary shift to facultative diapause for an insect behaving an obligatory diapause or developmental plasticity to alter voltinism productively for an insect with facultative diapause before shortening photoperiods inducing diapause. Inter-population evidence supporting the premise (theory) comes primarily from a model system with voltinism linked to thermal gradients across latitude. We examined the intra-population evidence in the field (47°24′ N, 123°68′ E) with *Ostrinia furnacalis*, one of the most destructive pests, on corn in Asia and Pacific islands. The species was univoltine in high latitudinal areas (≤46° N). Divergence of the diapause feature (obligatory and facultative) was observed within the field populations from 2016 to 2021. Warmer climates would provoke more facultative diapause individuals to initiate a second generation, which will significantly drive the population to evolve toward facultative diapause (multi-voltinism). Both divergent diapause and temperature must be considered for accurate prediction of phenology and population dynamics in ACB.

## 1. Introduction

Knowledge and interpretation of ecological adaptation have been considerably expanded by investigations of species evolving due to an expansion in favorable developmental seasons and/or to thermal-manipulated resources, either in the extended region or in their natural habitat in a warming climate. Critical studies, specific to insects, include those emphasizing species distributions (spreading out in new habitats, expanding overwintering range, feeding on different host plants) [1,2,3,4,5,6,7], an increase in annual generations (shortening life cycle, altering voltinism) [8,9,10,11,12,13,14], tri-trophic interactions (top-down and bottom-up) [15,16,17], etc. Limited empirical data and prediction models reveal that additional generations will be produced with a faster development rate of insects with a thermally driven longer growing season under a climate warming scenario, which will predictably lead to more frequent outbreaks and inevitably result in severe crop losses and/or increase the costs of control efforts [4,8,18,19,20]. It is easy to argue that shifting voltinism is realistic in insects existing in temperate zones with genetically programmed diapause (obligatory diapause) or photoperiodic calendar-regulated diapause (facultative diapause) natures, which are evolutionary life-history strategies to adapt to unfavorable environmental conditions [21,22,23]. Some species display the feature of vernalization, i.e., winter warming, which restrains development for springtime awakening, especially in areas with mild winters [24]. Vernalization effects, additionally, are not only species-specific [25] but also population-dependent [26,27]. Although intra-population variation may denote genetic variation, there is limited information concerning intra-population variation in response to climate warming. On the basis of the genotype, a temporally specific change in temperature may result in divergent effects in phenology. Winter chilling and spring warming lead to more synchronous adult emergence in many agricultural insect pests [25,27,28]. However, a significant gap in knowledge concerns such shifts in phenology that may alter voltinism and population dynamics.

The Asian corn borer (ACB), *Ostrinia furnacalis* (Guenée), is a notorious pest that damages corn, millet, sorghum, etc., in Asia and the Pacific islands. ACB has normally been categorized as a facultative diapause insect [29,30], regulated by a multigenetic or quantitative genetic makeup in response to day length [31]. Due to its plasticity in response to day length for development and diapause induction, populations of ACB vary geographically in voltinism, with increasing reproduction of generations toward the southern latitudes and a low altitude shift [32,33,34,35,36,37]. Spatial adaptation in ACB has usually been documented to the local light regime in accordance with the specific critical day length [29,33,35,38,39,40]. However, there have also been reports of the uni-, bi-, and multi-voltine ecotypes co-existing [41,42], even in high latitudinal (50°14’ N) areas [43]. Similar findings have also been reported in the European corn borer (ECB), *Ostrinia nubilalis,* in North America [44,45,46,47,48].

While day length is a predominant environmental cue regulating the onset of diapause in ACB [29,38], elevated temperature can override short day length, i.e., compensatory effect, and eliminate larval diapause incidence [34,49]. These have been reported in many other insects [50,51,52,53,54,55], including the ECB [48]. The diapause termination and development are driven by temperature, while the onset of diapause is primarily induced by the photoperiod. Time variation has been addressed for post-diapause development of co-existing uni- and bi-voltinism overwintering larvae in spring, i.e., days for developing to the pupa is shorter for bivoltine than univoltine larva [56,57]. Therefore, the activity of univoltine moths is later than bivoltine moths in spring. As a consequence of prolonged phenology, univoltine moth flight has usually been considered well synchronized with the later whorl stage of corn plant development, which results in more severe damage due to a much higher survival rate of larvae fed on this stage of corn plant [58].

Owing to its plasticity in ecological (i.e., diapause response) and biological (i.e., developmental rate) traits in response to both photoperiod and temperature, ACB is an ideal candidate for studying the mechanisms of voltinism alternation that influence population dynamics. Although the ACB moth has the potential for long-distance migration [59,60], historical data show that outbreaks of the ACB population were due to large, localized overwintering populations [58,61,62,63,64,65,66]. The univoltine of ACB has usually been described as severely damaging, although, as mentioned above, there have been reports of a second generation or a partial second generation south of the isotherm 2600 dd (degree-days) for accumulative active temperature (a minimum base threshold of 10 °C) in Northeast China [40,57,61,67,68,69,70,71]. This is changing, as more evidence shows that a second, and even a third partial generation, results in significant yield loss [43,72,73,74,75].

As a result of climate warming’s impact on agricultural systems, elevated temperatures have offered a longer growing season in Northeast China [76]; therefore, earlier planting and late maturity hybrids (high yield potential but greater thermal requirement) have been simultaneously adapted to extend the growing season [77,78,79]. Specifically, temperatures appear significantly elevated in winter and spring in Northeast China [80], which offers warmer conditions that favor shortening life cycles by ending larval diapause earlier and facilitating their post-diapause development. As the matter stands, the moth flight correspondingly occurs earlier. Moreover, early planting could provide available plants for egg laying. Climate warming could drive individuals that used to be univoltine to shift to bivoltinism due to the compensatory effect of elevated temperature. Meanwhile, late-maturity plants favor those bivoltine larvae completing their development to mature and entering diapause for overwintering. Altering the phenology of ACB will inevitably lead to the relevant adaptation of pest management tactics, such as the artificial release of *Trichogramma,* which has been widely implemented in China for decades as a basis for sustainable ACB management [81]. This study tested 3 hypotheses: (1) ACB has evolved a divergent diapause strategy; (2) the warming climate drives ACB to evolve toward dominance of multivotinism in a historically univoltine area; (3) severe damage by ACB is due to a high population of second and third generations. To test these hypotheses, we investigated the intra-population diapause in ACB, climate-induced population diapause shift, and phenological plasticity. We also developed a statistical model to simulate phenology and population fluctuation.

## 2. Materials and Methods

### 2.1. Experimental Location and Host Plant Cultivation

The experiment site was located at the Quanhetai village (47°24′ N, 123°68′ E), Qiqihar, Heilongjiang Province, China, a rainfed agriculture area planted with corn (50.1%), soybean (26.9%), and rice (17.5%) as major annual crops. Corn is normally planted in early May and harvested in early October. The experiment plot was 6 × 4 m^2^ and planted with 6 rows with 20 to 22 plants per row (Appendix A). Each plot was split into 2 subplots with 3 rows each. A commercial hybrid of sweet corn Cui Wang (Syngenta Co., Ltd., Qiqihar, China) was used as the host plant of ACB and was planted in 3 plots every 7 days from 10 May (the first planting date) to 21 June (the last planting date) every year from 2016 to 2021, which could offer similar developmental stages when the plant was infested with ACB neonates at several time intervals. A field-based nylon screen cage (6 × 4 × 2.5 m^3^) was deployed for each plot to protect corn plants from infestation by natural lepidopteran pests such as ACB, oriental armyworm, and *Mythimna separata* (Walker).

### 2.2. Assessment of ACB Voltinism

#### 2.2.1. ACB Moth Collection

A high-pressure mercury lamp (500 W) was set up in the Experiment Station of Qiqihar Sub-Academy of Heilongjiang Academy of Agricultural Sciences, Qiqihar, from 20 May to 30 September from 2016 to 2021. The light was turned on at 18:00–23:00 every day except on strong wind or rainy days. A white curtain was placed near the lamp for the moth landing. ACB moths were collected and transferred into an aluminum screen cage, which was then brought to the laboratory for egg laying under the conditions of 28 °C, 16:8 h (L:D) photoperiod, and 80% relative humidity. Moths collected from 5 consecutive days were pooled as a time interval collection. Their egg masses were pooled and placed in an incubator. After hatching, the neonates were used for field infection.

#### 2.2.2. ACB Infestation and Diapause Survey

The field infestation with neonates was generally performed every five days from 10 June to 31 July. Three plots were designed for each time, but only one of two subplots was infested per plot; the typical field infestation technique [82] was used to transfer neonates into whorl leaves of the V6-7 corn plant. Each plant was infested with ca. 60 neonates (<12 h). All infested plants were dissected 45–50 days after infestation. If a plant in a neighbor row showed clear leaf and stalk damage, it was also dissected. The numbers of pupae and larvae were recorded for each subplot and then transferred to the plants in the neighbor subplot. All plants were visually checked for ACB egg masses (next generation) before dissection. If any existed, they were also transferred into plants in the neighboring subplot. At the end of the season, all plants in the other subplot were dissected to record the number of larvae and pupae.

#### 2.2.3. Modeling Incidence of Diapause

The onset of diapause was induced by photoperiod, which is alterable with temperature. Therefore, we used data on diapause dynamics to model the proportion of first-generation larvae that developed into diapause, *P_d_*, over the day length, *d_l_*, of the date for field infestation with neonates across 6 years using a modified logistic model by reversing the signs of parameter *k* (i.e., rate of increase) and *b* (i.e., lag) to allow the proportion to be modeled over day length (a decreasing continuous variable) of the date for field infestation with neonate larvae [83].
(1)$$P_{d}={\left(1+\exp\left(kd_{l}-b\right)\right)}^{-1}$$

Nonlinear model fitting was performed using Regression-Nonlinear of SPSS Statistics 20.0 (IBM Corp, New York, NY, USA, 2011). Day length data for Qiqihar were obtained through the website (https://richurimo.bmcx.com/qiqihaer__richurimo/?ivk_sa=1024320u) (assessed on 8 May 2022).

### 2.3. Phenology

#### 2.3.1. Modeling First Moth Flights

In a laboratory experiment (Liu et al., unpublished data), the thermal requirement is ca. 1036 dd (with a minimum base temperature threshold of 6.4 °C) for 90% of the overwintered larvae (851 larvae) developing to pupation at 28 °C, 70% relative humidity (RH) with 16 h daylength. For pupa developing to adult as well as preoviposition, it needs 115.4 dd (with a minimum base temperature threshold of 11.8 °C) [84] and 22.5 dd (with a minimum base temperature threshold of 10.0 °C) [85]. Based on these thermal data and dynamics of light trap catches, we estimated the timing of the first moth flight, which was probably initially from 21 to 29 May and ended from 15 to 23 July. According to the best fit, found by comparing *R*^2^ values among logical, linear, and Gompertz models, a modified Gompertz model was finally used to model the proportion of first moth flight, *f*_1_, over effective accumulative temperature (accumulated degree-days), *T_d_*_._
(2)$$f_{1}=\exp\left(-exp\left(-xT_{d}+b\right)\right)$$

In which *x* and *b* are the rate of increase and lag, respectively [83]. Nonlinear model fitting was performed using Regression-Nonlinear of SPSS Statistics 20.0 (IBM Corp, New York, NY, USA, 2011). Daily degree-days from 1 March to 30 September for each year were estimated using a degree-day calculator developed by the University of California (http://ipm.ucanr.edu/WEATHER/index.html) (accessed on 10 May 2022) which is based on the single sine-wave method with a horizontal cut-off to calculate effective heat unit accumulations. The minimum base threshold for larval development is 6.4 °C [84]. Temperature data for Qiqihar were obtained through the website (http://tianqi.2345.com/wea_history/) (assessed on 8 May 2022).

#### 2.3.2. Modelling and Predicting Second and Third Flights

Both the cumulative temperature and photoperiod govern the phenology of multivoltine ACB in distinct regions, particularly in Northeast China; the former mediates development, and the latter regulates the onset of diapause. Therefore, the timing of second-generation moth emergence was estimated over a degree-day scale based on the empirical *T*_dd_ requirement of adult-to-adult development from the date of first flight. The *T*_aa_ for key phenological events were preoviposition, 22.5; egg 44.7; larva, 452.1; and pupa, 115.4 dd, respectively [84,85], and a total was 634.7 dd. The minimum base thresholds for pre-ovipositional, egg, larval, and pupal developments are 10, 13.5, 6.4, and 11.8 °C, respectively. Meanwhile, the percentage of offspring individuals developing to the second generation was adjusted with date-specific diapause incidence, estimated from our models of daylength–diapause response derived from the experiment of assessment of ACB voltinism each year. The proportion of second moth flight was calculated based on the predicted second-generation moth emergence and trap catches, respectively. In order to yield the best fit, either the modified Gompertz model (Equation (2)) or logical model (Equation (3)) was used to model the proportion of second moth flight, *f*_2_, over *T*_dd_ by comparing *R*^2^ values.
(3)$$f_{2}={\left(1+\exp\left(-xT_{d}+b\right)\right)}^{-1}$$

## 3. Results

### 3.1. Intra-Population Variation of Diapause

For all 6 years of field trials of assessing ACB voltinism, there were 51.3–62.8% of first-generation larvae (neonates hatched from eggs laid by first-flight moths) developing to pupa–adult and producing offspring in summer, which demonstrated facultative diapause or non-diapause (bi- or multi-voltinism) and was going to enter diapause or to die at the end of the season (Appendix A), whereas others would directly enter diapause infestation during summer solstice time (univoltinism), i.e., displaying obligatory diapause.

### 3.2. Climate-Induced Population Diapause Shift

The dynamics of larval diapause incidence for the first generation demonstrated a similar trend across 6 years, i.e., the incidence of diapause was proportional to post-summer solstice day lengths (Figure 1). However, the difference in critical day length for the onset of diapause was observed over the years. The model estimated the critical day length for 50% diapause incidence was in the range of 15 h/30 min to 15 h/42 min, which corresponded to the dates of 11 to 17 July on the calendar. Obviously, the critical day length was longer in 2016–2019 than in 2020–2021. Coincidently, discrepancies were observed in temperatures from 8 to 21 July over six years, which reflected that the value of dd during 8–21 July was higher in 2020–2021 than in 2016–2019 (Figure 2). In other words, an increase as small as 1.2 °C in daily temperature from 8 to 21 July led to a shift from high larval diapausing (50% in 2016) to pupation (81% in 2020).

### 3.3. Spring Moth Emergence

From 2016 to 2021, the first trap catches of the spring emergence of the overwintered generation varied from 21 to 29 May. On a calendar timescale, the timing (beginning) of first flight (i.e., first-generation moth emergence) peak or median of peaks (i.e., 50% first-generation trap catches) varied from 12 to 27 June across 6 years. However, on the basis of accumulated degree-days (dd) from 1 March with a minimum base threshold of 6.4 °C for larval development, the nonlinear regression model estimated that 50% of first-generation trap catches were 739, 720, 706, 763, 847.5, and 787 dd for 2016–2021, respectively (Figure 3), which was in correspondence with the dates of 27, 27, 17, and 26 June, 4 July and 30 June on a calendar timescale. The empirical dd for 50% of first-generation trap catches was demonstrably able to classify into two levels in synchrony of emergence, i.e., a low level of thermal requirement for 50% of overwintering larva-to-adult development (2016–2019) and a high level (2020 and 2021), respectively.

### 3.4. Phenological Plasticity

The timing of moth flight occurred from late May to the end of September over the course of the 2016–2021 maize growing seasons (Appendix A). Totals of trap catches were 7542, 7299, 11,162, 4614, 1701, and 2773 from 2016 to 2021, respectively. Graphically, the dynamics of moth appearance illustrated primarily two or three visually distinguishable peak periods each year, i.e., there were two or even three discernible emergences within the maize growing season in Qiqihar. Based on the thermal requirements for each key phenological event and first-flight fluctuation linked with the incidence of larval diapause dynamics in each year, second-moth flight dynamics were predicted for each year (Figure 3). The predicted dd for 50% of second moth emergence were 1410 (2016), 1424 (2018), 1449 (2019), 1520 (2020), and 1433 (2021) dd, respectively. In contrast, the estimated dd for 50% of second-flight trap catches were 1609, 1640, 1666, 1595, 1515, and 1448 dd from 2016 to 2021.

In comparing the curves of second-generation moth emergence from first-flight prediction and second-flight trap catches, discrepancies were observed between the two models for 2016–2019. The estimated differences between 50% moth emergence and the corresponding percentile for trap catches were 145 to 242 dd, respectively, which roughly corresponded to a total dd of 10 to 17 days in midsummer. However, this pattern was not observed in 2020 and 2021. The predicted dd of 50% moth emergence was nearly similar to or even higher than that for trap catches.

When second-flight moths emerged before 9–15 August 2016–2021, their offspring could develop to maturity and enter diapause at the end of the season (Figure 4), whereas those emerging after this date were unable to complete larval development and would die in winter. Based on this, 89.0–90.7% of bivoltine individuals would complete two full generations in warmer years (2016 and 2018), whereas it would decline to 51.1–79.8% in cooler years (2019–2021) (Appendix A). The warmer was in a year, especially during the months of August and September (Appendix A), the later second-flight moths emerged, and their offspring could complete larval development at the end of the season. In consequence, there were 40.9–68.5% of the populations in the second generation able to complete larval development in 2016–2021.

The estimated dates of second-generation moth emergence were 11–19 July across 2016–2021 (Figure 4). In addition, the dates of first-generation moth emergence at 90% population level were 15–23 July. Obviously, there was a time overlap between first- and second-generation moth occurrence (Figure 3). Since moths were emerging on 11 July to 1 August, their offspring could complete development from egg to adult before the end of the season. Therefore, the nonexistence of a partial third generation would be rare.

## 4. Discussion

Historically, it is considered that ACB could have one or one and a second incomplete generation within a year in an area with the effective accumulated temperature ≤2600 dd, which is mainly based on the field survey for deposited egg masses on corn plants during the season [40,61,71]. The dynamics of egg-laying showed a single peak timing curve in Daqing (46°04′ N, 124°81′ E), Heilongjiang Province, which was initiated from 9 June, gradually increased, reached peak timing during 15–23 July, and ended on 16 August [86]. Therefore, these studies suggest that the corn plant damage is mainly caused by the first-generation insect. Meanwhile, if the timing of egg-laying activities is initially from the end of June to early July and the generation peak occurs from middle- to late-July, serious damage or an outbreak result. Similar results have also been reported even in the more southern population in Gongzhuling (43°31′ N, 124°50′ E, elevation 216 m), Jilin Province [57,64,70,87]. However, these studies suffer from inappropriate interpretation (only based on one highest peak appearance during the growing season) for the dynamics of egg deposition. The natural resources (host, temperature, humidity, photoperiod, etc.) are available for ACB to reproduce two generations in Qiqihar and more south (Appendix A). According to the accumulative temperature requirement for development [84,85], predicted egg deposition timing of first- and second-moth flights are approximately from 23 May to 25 July and 12 July to 29 September, respectively. Obviously, the egg-laying period of first- and second-moth flights overlap (Figure 3). Together with the low population density of the first-generation, the timing of generational peaks (especially first-generation) of egg laying is visually indistinguishable through a line graph illustrating seasonal fluctuation of observed egg deposition on a calendar timescale. It is reported that the egg-laying period of the ECB in the central corn belt during the second-moth flight is represented by a 20-day triangle, with the peak egg deposition 10 days after the first eggs are laid [88]. It is cooler in Qiqihar; therefore, the egg-laying period could be longer.

Our study has shown that ACB moth population dynamics display mainly two peak timings within a year as measured using a light trap in Qiqihar. The moth population is much higher in the second peak timing than in the first peak timing (62.1–78.5% of the total yearly population vs. 21.5–37.9%). There were 51.3% and more offspring of first-emergence moths developing bivoltinism or incomplete bivoltinism. These results revealed that ACB has mostly performed bivoltinism in Qiqihar. The second-generation population would lead to more serious damage to corn plants compared to the first generation.

These findings are contradicted by the report of Hu [89], who concluded that ACB could only complete one generation per year in Suihua (46°42′ N, 127°43′ E, elevation 180 m) and north. It is difficult to make a direct comparison of the data between the two studies due to the inappropriate interpretation of light trap catches in Hu’s study [89], i.e., the adult emergence from overwintering diapause larva was initially from 16 June, then gradually increase to a peak timing of 28–30 July, and ended on 13 August. Apart from the inappropriate interpretation of light trap catches, Hu’s [89] study further suffers from the late initial trapping time. The initial light trap catches are on 8 June or earlier in the more northern population in Nenjiang (49°40′ N, 125°30′ E), Heilongjiang Province [90]. In comparison, our study started trapping in the middle of May, and the first catch of moths was on 21 May.

Our results for intra-population variation in voltinism are in accordance with other studies [38,41,42,43,67]. Despite not being aware of genetically predetermined diapause (obligatory diapause) involvement for univoltinism, Lu et al. [42] conclude that univoltine and bivoltine individuals respond to different photoperiods for inducing the onset of diapause and the critical day length is longer for univoltine (14 h/30 min) than for bivoltine (13 h/40 min). However, univoltinism has been observed in long day conditions either in the field (from June to July) [41,43] or in the laboratory (L:D 16:8 h, 28 °C) [31,38], suggesting that the population consists of individuals with genetically predetermined diapause (obligatory diapause) besides photoperiodic programmed diapause (facultative diapause), i.e., sympatric voltinism (uni- and bi-voltine) strains. This agrees with our findings. Similar findings have also been reported in ECB [44,91].

Apart from the unique univoltine strain in the population, photoperiodic programmed facultative individuals were dominant in the previous studies [31,38]. Despite population differences, our findings matched theirs very closely. In addition, temperature-compensated photoperiodic programming development and matured larval diapause have been reported in diverse geographic populations [31,32,34,38,92]. In this study, an increase as little as 16 dd in accumulative temperature during 8–21 July led to 12 min shorter critical day length, suggesting that a higher percentage of bivoltine and/or incomplete bivoltine individuals will be introduced into the population under a climate warming scenario.

Insect diapause response is assessed based on the response of the population [93], although it is an individual binary trait that could be observed only as a single response from an individual in its lifetime. In this study, the timing of the last trap catches (adults) was in late September (25 September 2017, 21 September 2019). Thus, their larvae should develop in August. While day length is shortened from 14.9 h on 1 August to 13.4 h on 31 August, and the daily high and low temperatures decline from 28 and 20 to 23.15 °C, which are typically environmental factors inducing the onset of larva developing diapause [32]. Obviously, there might be multivoltine (non-diapause) individuals in the population. This agrees with the finding in other ACB and ECB studies [31,38,91].

Overall, our light trap catches results show that ACB moth flight phenology had large discrepancies between years in relation to the timing of the first catch, median peaks, and the last catch of moths on a calendar timescale. Additionally, it was with early initial (i.e., 21 May or even earlier) and late end (i.e., 25 September) in warm spring and summer years, while it was an inverse pattern in colder years. These results suggest that elevated temperatures in spring and summer will speed up insect development under an ongoing climate change scenario, which may lead to an additional voltinism. This agrees with findings in other ACB studies [43,72].

Although the phenology of the moths from light trap catches varies between years on a calendar timescale, the timing of generation peaks or median of peaks is more similar on a degree-day scale across 4 years (2016 to 2019). A similar finding is also reported in the ECB [88]. This supports the hypothesis of degree-day models for predicting insect life-cycle stage based on temperature-dependent development time, i.e., the proportion of generation moth flight over accumulated degree-days was well fitted a modified Gompertz model [83].

In comparing models predicting second-moth flight based on the spring emergence incorporating the incidence of diapause with the corresponding photoperiod on that date and the light trap catches, there are discrepancies in the timing of generation peaks or median peaks between them. The timing of prediction at the 50% moth emergence was 10–17 days earlier than the corresponding percentile for light trap catches for 4 years (2016–2019). This time interval is due to ACB life expectancy. On average, male and female moths could survive 8–17 days at temperatures of 16–24 °C and ≥70% RH under laboratory conditions [30,94,95].

However, it was divergent from 2020 to 2021. The timing of spring emergence for 50% of the overwintered generation was discernibly late in comparison with the other 4 years. Meanwhile, the population levels declined extraordinarily from 2020 to 2021. One of the reasons might be a few farmers planting Bt maize without permission. Prolonged development time has been reported for Bt-resistant insects [96], including a lab-selected Cry1Ab-resistant ACB strain [97].

## 5. Conclusions

ACB has evolved divergent diapause features (obligatory, facultative, and non-diapause) to survive in winter (unfavorable time of the year) and to develop and reproduce in order to maximize its population in the growing season (favorable time of the year). A warmer climate stimulates more facultative diapause and/or non-diapause individuals producing additional generations, which results in a higher proportion of facultative diapause individuals in the population. Consequently, this will drive the population to evolve toward facultative diapause. Meanwhile, the population, which used to be univoltine, is becoming bivoltine or even trivoltine in response to the ongoing warming climate scenario in high latitudinal areas. The phenological plasticity of the population depends on a combination of intra-population diapause variation and temperatures; thus, both must be taken into account for accurate predictions of phenology.

## Figures and Tables

**Figure 1 biology-12-00762-f001:**
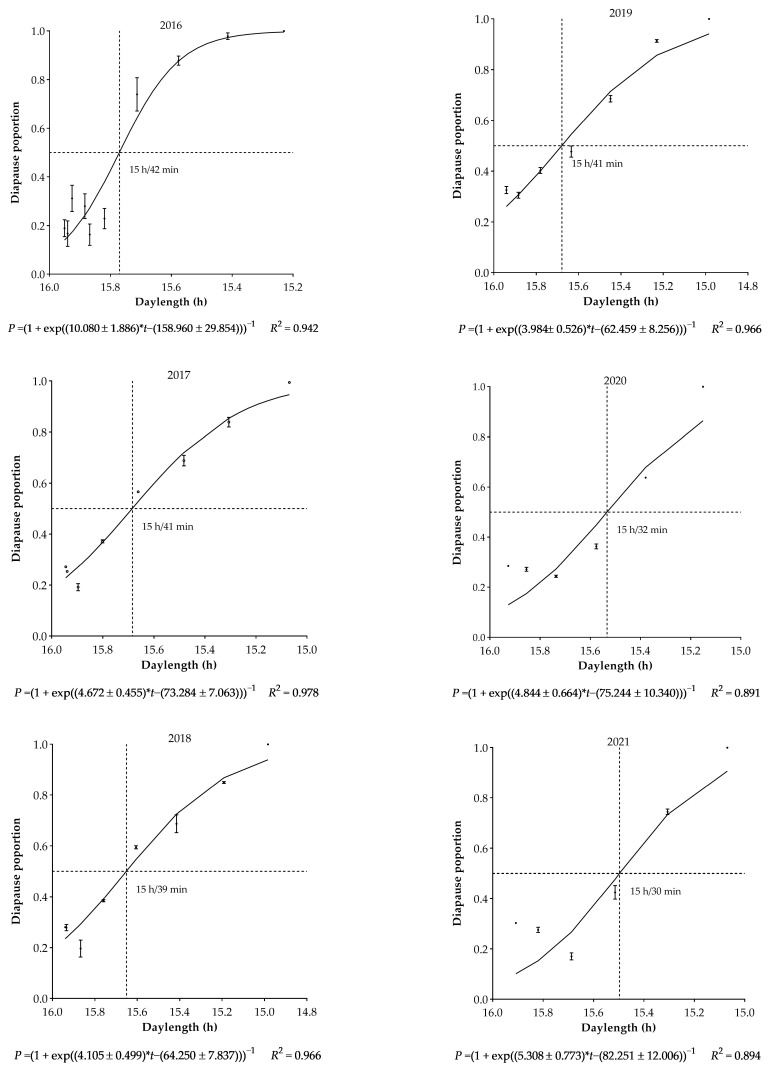
Incidence of larval diapause in first-generation *Ostrinia furnacalis* in the field in Qiqihar, Heilongjiang Province.

**Figure 2 biology-12-00762-f002:**
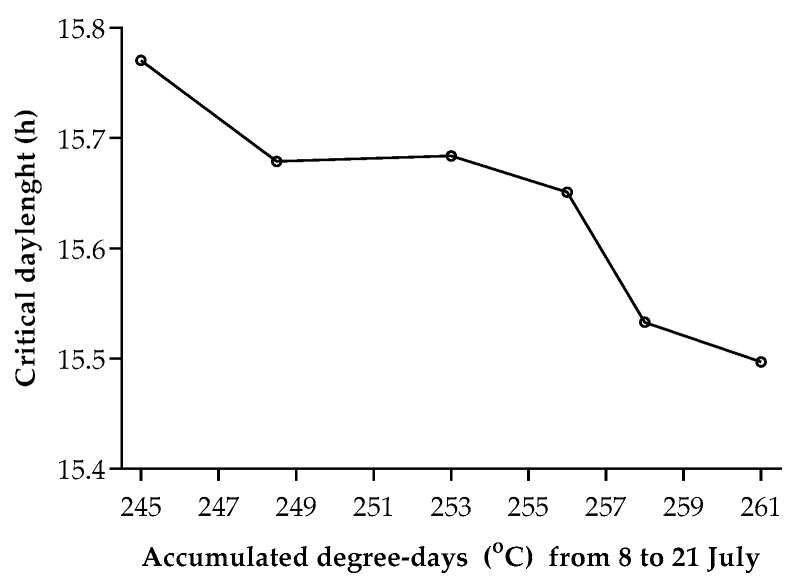
The compensatory effect of temperature on short day length inducing diapause in *Ostrinia furnacalis*.

**Figure 3 biology-12-00762-f003:**
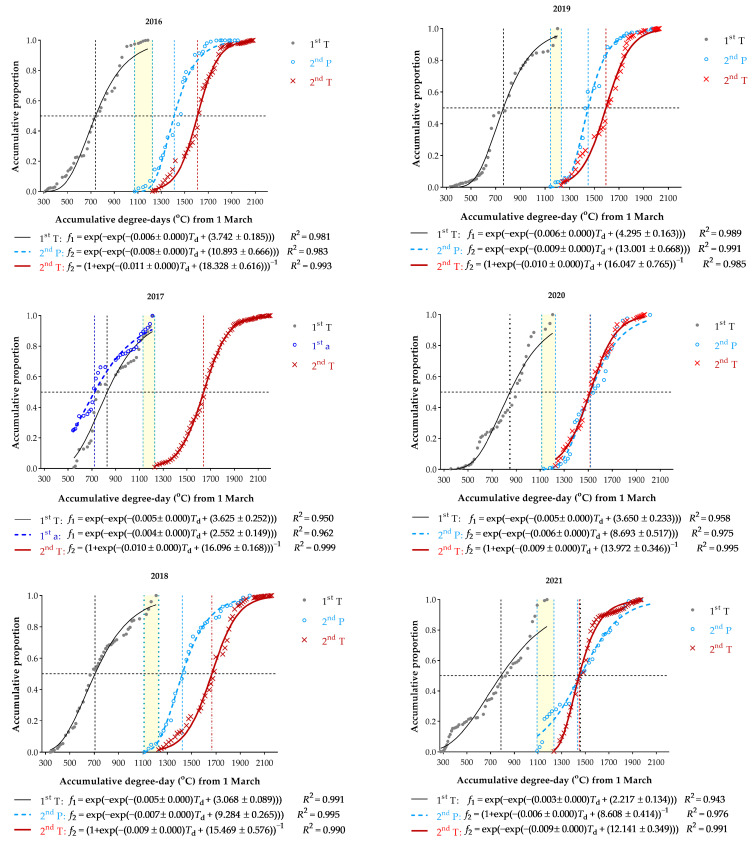
Cumulative proportion of light trap catches of *Ostrinia furnacalis* over accumulated degree-days across 6 years in Qiqihar. 1st T: first flight trap catches; 2nd P: second emergence prediction; 2nd T: second flight trap catches; 1st a: adjusted first trap catches; because the light trap was started late (on 10 June 2017), we estimated there should have been ca. 25% adult emergence from diapausing larvae before 10 June 2017, according to 2016, 2018, and 2019 trap catches; thus, the total of first flight trap catches (*T*) was estimated using the sum of trap catches (∑*T*_i_) from 10 June to 15 July as 0.75 proportion, i.e., $\frac{\sum T_{i}}{T}=0.75$. The accumulative proportion (*P*_i_) of moths captured in the light trap over the course of first flight (10 June to 15 July) was calculated as: $P_{i}=0.25+\sum \frac{T_{i}}{T}$, in which *T*_i_ is the number of traps catches each day.

**Figure 4 biology-12-00762-f004:**
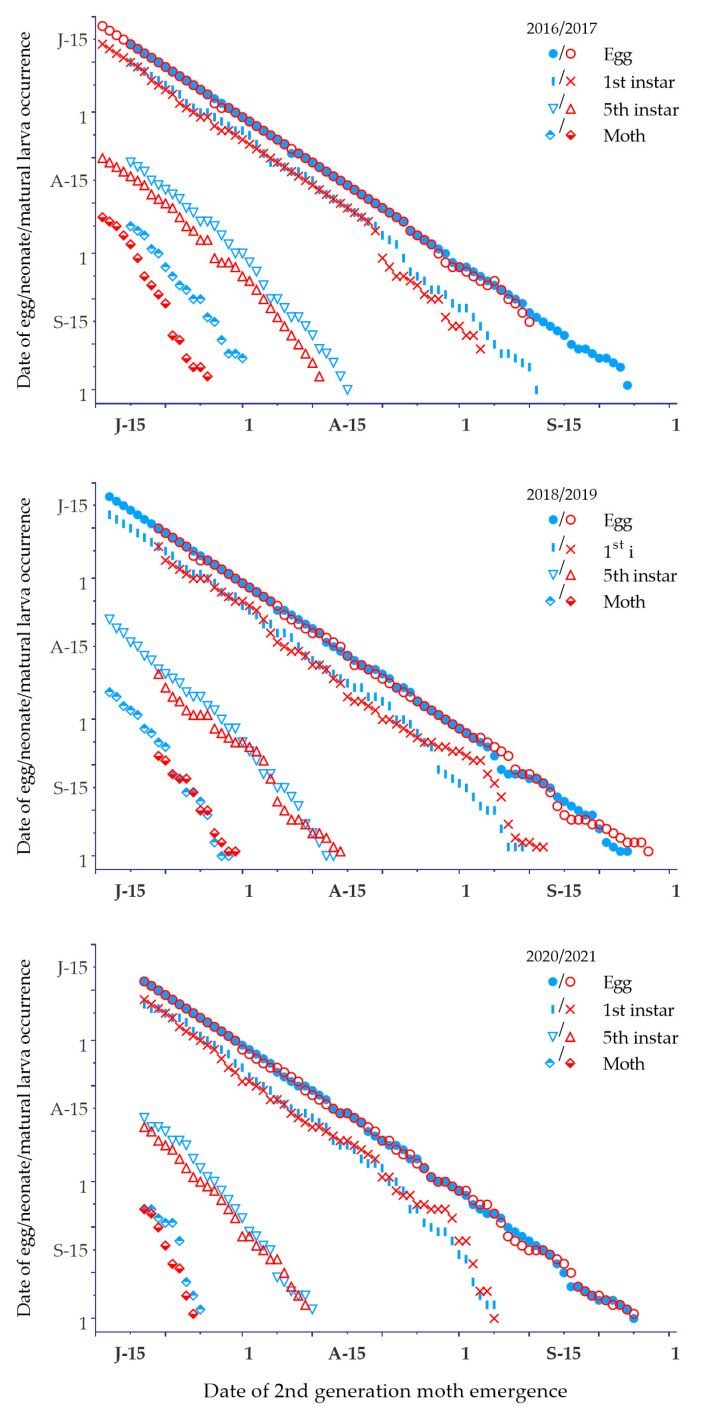
Predicted date of second- and third-generation egg, neonate, mature larva, and adult occurrence of *Ostrinia furnacalis* based on the accumulative temperature requirement from second adult emergence in Qiqihar.

## Data Availability

Not applicable.

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
