# Peer review of "Evolutionary Shift of Insect Diapause Strategy in a Warming Climate: An Intra-Population Evidence from Asian Corn Borer"

_biology, 2023, doi:10.3390/biology12060762_

Round 1

Reviewer 1 Report

The manuscript Biology 2327950 is an interesting study unraveling the plasticity of the ACB phenology. This study has important implications, which I found highly suitable for the journal.

Still, there are four things that might be considered by the authors to improve the readability and quality of the manuscript.

1. Since the manuscript is based on experimental investigation, it would be much better to set up and test the hypotheses. The introduction gives a nice and complete overview of the state of the art knowledge; however, the ending is not impressive. I was waiting for some testable hypotheses and found a very general single sentence about the rest of the paper.

2. The first paragraph of the experimental design is not immediately clear. After reading of 3 times, I am not still sure the structure of the experiment. I suggest improving the description, or you may even put an image showing the experimental set-up.

3. I missed some statistical considerations. This reflects the first issue. Would be nice to get testeble hypothesis and provide some significance levels of your findings

3. Finally, there are a number of minor language issues that should be corrected.

regards

Author Response

  1. Since the manuscript is based on experimental investigation, it would be much better to set up and test the hypotheses. The introduction gives a nice and complete overview of the state of the art knowledge; however, the ending is not impressive. I was waiting for some testable hypotheses and found a very general single sentence about the rest of the paper.

Response: Thank you for your suggestion, we added two testable hypotheses in the introduction section.

  1. The first paragraph of the experimental design is not immediately clear. After reading of 3 times, I am not still sure the structure of the experiment. I suggest improving the description, or you may even put an image showing the experimental set-up.

Response: We are sorry for the unclear description. We added a new Fig. S1 to show the experiment design.

  1. I missed some statistical considerations. This reflects the first issue. Would be nice to get testable hypothesis and provide some significance levels of your findings

Response: Thank you for your suggestion, we added the hypothesis and highlight the findings. Line 105-109

  1. Finally, there are a number of minor language issues that should be corrected.

Response: We checked the manuscript throughout and corrected all the mistakes.

Reviewer 2 Report

Wang et al present field evidence for divergence in the diapause strategy of the Asian corn borer, which they link to warmer growing conditions. The study is impressive in its temporal extent and there are significant findings in the results. The introduction adequately sets out the research questions and the discussion adequately presents the findings in the context of other work. The methods and results are well presented though the manuscript as a whole would benefit from editing by a native English speaker as there are some sentences that are unclear in the present form. I do not have any other comments to make on the manuscript other than to question the choice of colours in Figure 3 and Figure 4 (red and green) which would be difficult for colour-blind people to distringuish.

Author Response

Thank you for the comments and suggestions. We modified the Figure 3 and Figure 4.